# Impact of Storm Characteristics on Infiltration Dynamics in Sponge Cities Using SWMM

Yuanyuan Yang *, Zijian Shao, Xiaoyan Xu and Dengfeng Liu 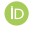

State Key Laboratory of Eco-Hydraulics in Northwest Arid Region of China, Xi'an University of Technology, Xi'an 710048, China; 2220421325@stu.xaut.edu.cn (Z.S.); liudf@xaut.edu.cn (D.L.)
* Correspondence: yuanyuanyang@xaut.edu.cn

**Abstract:** Effective stormwater management in urban areas requires enhancing the permeability of underlying surfaces. However, the impact of storm characteristics on infiltration processes in sponge cities remains insufficiently explored. This study uses the Horton method within the storm water management model to investigate how uniform and Chicago storm parameters affect infiltration rates. Our findings provide valuable insights: (1) Increasing porous pavement area proportionally reduces subarea sizes within subcatchments, and infiltration rates of porous pavements are supply-controlled. (2) Uniform storms result in consistent initial infiltration rates across pervious areas, subcatchments, and the entire catchment. The duration of this stable state decreases with higher return periods. Catchment infiltration volumes exhibit linear growth with greater storm intensities ($R$-squared = 0.999). (3) Peak infiltration rates and moments for pervious areas, subcatchments, and the overall catchment exhibit correlations with both the return period and the time-to-peak coefficient, with correlation coefficients ranging from −0.9914 to 0.9986 and $p$-values ranging from 0.0334 to 0.6923. This study quantifies the influence of design storm parameters on infiltration, providing valuable insights for stormwater infrastructure design and urban stormwater control.

**Keywords:** Chicago storm; Horton; porous pavement; return period; time-to-peak coefficient

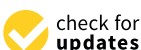



## 1. Introduction

Infiltration, the process of water movement from the surface into the soil and subsurface driven by gravity and soil capillarity, plays a vital role in the redistribution of water resources and significantly impacts various hydrologic processes (e.g., runoff generation [1], groundwater recharge [2]) in urban catchments. Accurate simulation of infiltration is a subject of interest in hydrological modeling, particularly in the context of rainfall–runoff models. Understanding infiltration dynamics and mechanisms in sponge cities, where low impact development facilities (LIDs) are employed, holds substantial potential to enhance urban stormwater modeling and management [3,4]. Sponge cities typically employ infiltration-based and retention-based strategies. Therefore, gaining insight into infiltration characteristics is pivotal for comprehending their hydrological responses and achieving effective stormwater control.

Directly measuring infiltration at a large-scale field is time-consuming, costly, and subject to significant spatial and temporal variability. Consequently, numerous theoretical and empirical infiltration models have been developed for indirect estimation [5,6]. Infiltration models can be categorized into two types [7]: physically-based equations such as Horton [8–10], Green–Ampt [11], Soil Conservation Service [12], Swartzendruber [13], Kostiakov, Kostiakov–Lewis, and Philip; and empirical and data-driven methods including artificial neural networks [14], support vector machines [15], random-forest models [16], and Gene Expression Programming [17].

Theoretically, the process of soil infiltration is governed by the Richards equation. The equation is a highly nonlinear partial differential equation and challenging to solve. So, the

Storm Water Management Model (SWMM) [18] is used in this study, which employs various basic algebraic infiltration models that represent the general dependency of infiltration capacity on soil properties and the volume of water previously infiltrated during a storm event. There is no consensus over the optimal algebraic infiltration model; that is, the physically-based infiltration models show varying levels of effectiveness and applicability. For example, the Horton and Green–Ampt methods underperform the modified Philip's model [19]. Therefore, SWMM allows the user to select from five of the most popular models: the Horton method, the modified Horton method, the Green–Ampt method, the modified Green–Ampt method, and the Curve Number method.

The Horton method in SWMM is chosen in this paper to synthetically produce infiltration data on urban permeable surfaces for three reasons. First, the Horton method, as the default infiltration model in SWMM, is widely used and offers reliable predictability for estimating rainwater infiltration into the upper soil zone [20]. For example, the Horton model outperforms Kostiakov and Philip models in built-up surfaces [21] and in semiarid regions; the Horton model outperforms the Curve Number method for grass soils [22]. Second, the Horton model often fits experiment data well [23] and has a few parameters that can be obtained with easy monitoring [24]; in contrast, the fitting accuracy of other models requires advanced field investigations [25]; for example, the performance of Green–Ampt model is considerably affected by the monitoring area and hydraulic gradients [26]. Third, our study site is located in a semiarid region [27] where storms predominantly result in infiltration-excess (or Hortonian) overland flow rather than saturation overland flow [28].

Infiltration capacity and rate on urban permeable surfaces are influenced by soil conditions and properties, such as moisture content [29–31] and structure [32]. Additionally, storm characteristics play a significant role [33]. The Horton method is susceptible to rainfall intensity [34] and temporal distribution [35,36] in semiarid regions. Our previous studies have shown that the performance of LIDs generally declines with less frequent and more intense storms [37], and the time-to-peak coefficient of rainfalls impacts runoffs in sponge cities [38].

This study's core focus and novelty reside in investigating how storm parameters influence infiltration rates. We employ both the Horton and Green–Ampt methods within SWMM [39]. Notably, the Horton method is applied for permeable surfaces, while the Green–Ampt method is utilized for modeling LIDs (i.e., porous pavements) in sponge cities. Our findings unveil the profound influence of storm characteristics on infiltration processes. These results underscore the potential benefits of augmenting porous pavements and gaining comprehensive insights into infiltration behavior under various storm scenarios, ultimately enhancing urban stormwater management practices.

## 2. Study Area and Data

Our research focuses on the WR8 site ($8.5 \times 10^5$ m$^2$, Figure 1), an urban drainage basin in the experimental sponge city of Fengxi, China, designated as a UNESCO Ecohydrology demonstration site [40]. The climate of WR8 falls under the warm temperate semiarid continental monsoon classification, characterized by pronounced seasonal variations in temperature and humidity. Over a year, the region receives a total of 1983.4 sunshine hours, with an average annual temperature of 13.6 °C. Notably, July exhibits the highest temperatures, averaging 26.8 °C, while January is the coldest month, with an average temperature of −0.5 °C. Precipitation displays substantial interannual fluctuations, with values notably surpassing evaporation. This study area experiences an average annual precipitation of 552.0 mm (averaged from 1981 to 2016, excluding 1986 and 2011), with a notable concentration of 50–60% falling between July and September [41]. Additionally, the average wind speed registers at 1.5 m/s.

WR8 features a prominent loess layer spanning elevations from 380.5 to 384.3 m above sea level. The soil composition predominantly comprises loamy clay, characterized by a compact structure with a yellowish-brown appearance, sparsely inhabited by plant roots,

and punctuated by needle-shaped holes and insect burrows. The groundwater table depth typically ranges from 10 to 20 m. The land use in the WR8 site encompasses diverse categories, encompassing parks and green spaces, residential lands, transportation lands, educational lands, industrial lands, and undeveloped areas. Stormwater finds its way to the Fenghe River via a designated outfall.

The drainage system in WR8 was mainly designed to accommodate storms with 1- or 2-year return intervals before 2014, resulting in frequent waterlogging events due to inadequate drainage capacity. Since then, the region has implemented LID-based stormwater management technology to mitigate storm-related problems. Numerous porous pavements (PP or permeable pavements, Figure 2) [42,43] have been implemented, covering 134,522 m$^2$, accounting for 15.8% of the total catchment area. PP has a stratified system including surface, pavement, storage, and underdrain components. Stormwater permeates each layer vertically. If the drainage rate exceeds the capacity of the underdrain, the water level will rise until it reaches the ground's surface, resulting in runoff. The water in PP can leave the bottom via percolation and evapotranspiration and be routed to a sewer junction or pervious area via the drain.

Crucial data, including precipitation data, land use, elevation information, details about storm-related facilities, and surface and pipe flow data, were provided by the Fengxi New City Management Committee [44]. For analytical purposes, the WR8 site was divided into nine subcatchments, 21 nodes, and one outfall in SWMM.

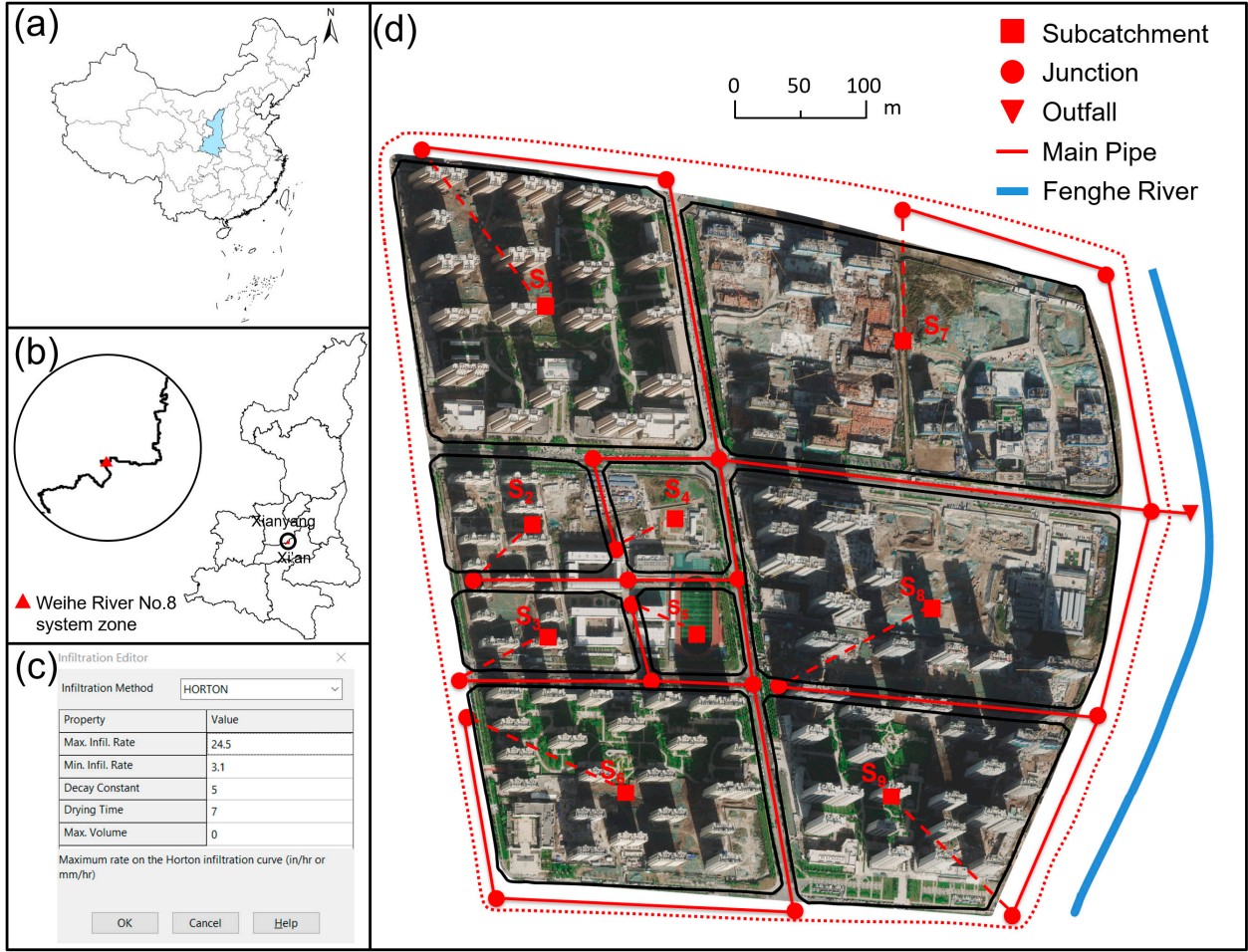

**Figure 1.** Study area (adapted with permission from Yang et al., 2023 [45]. 2023, Elsevier). (**a**) Shannxi Province, China; (**b**) Weihe River No. 8 system zone (WR8), Fengxi New City; (**c**) Infiltration editor in storm water management model (SWMM); (**d**) Aerial photograph of WR8 overlapped with SWMM generations.

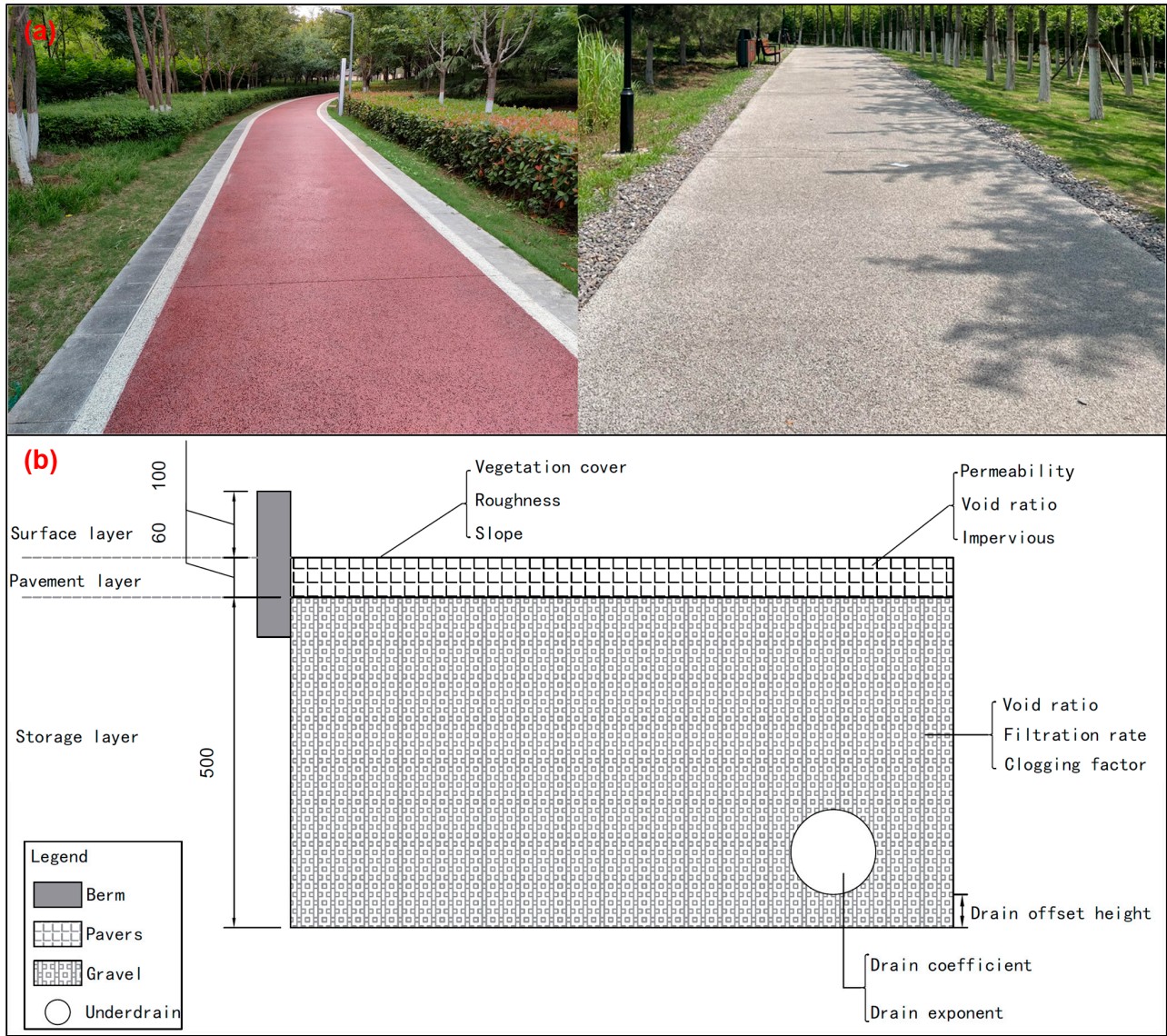

**Figure 2.** Porous pavements in study area. (**a**) Photograph. (**b**) Profile.

## 3. Methods

To analyze the influence of storm parameters on infiltration dynamics, we have established a framework utilizing the SWMM engine in Visual Studio 2022 for conducting stormwater simulations (Figure 3). In this framework, MATLAB is employed for storm design. The framework comprises four main components: (1) Designing uniform and Chicago storms with various parameter values. (2) Executing SWMM simulations to compute the time series of infiltration rates in each subcatchment and their corresponding subareas. (3) Calculating infiltration statistics, including peak rate, peak time, and volume. (4) Assessing the impact on the infiltration statistics.

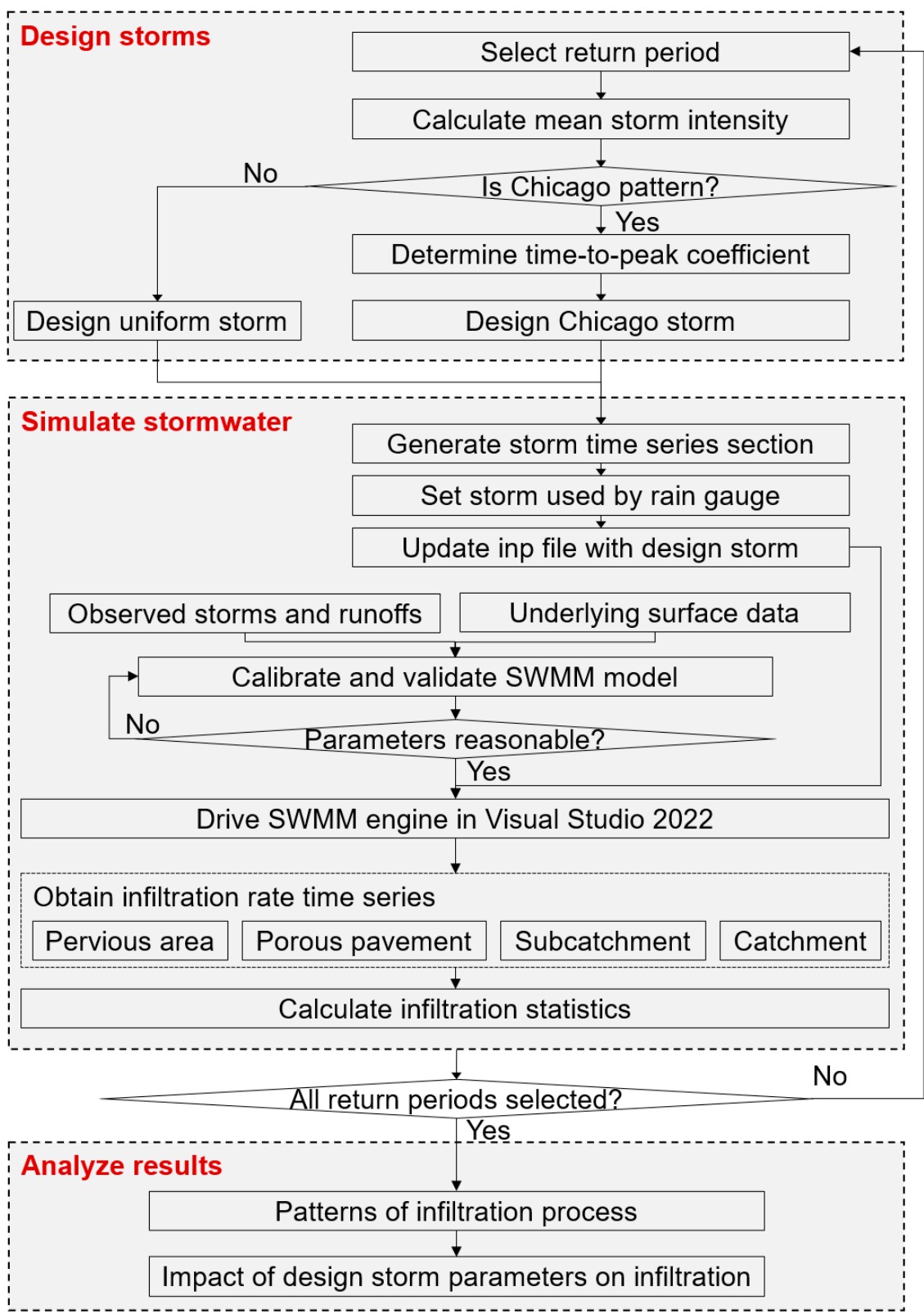

**Figure 3.** Analytical framework.

*3.1. Design Storm*

The uniform storms were designed according to the duration–intensity–frequency approach:

$$q = \frac{a(1 + c\lg T)}{(d + b)^n} \tag{1}$$

where $q$ is the average intensity, mm/min; $a$ is the storm coefficient; $c$ is the coefficient of variation; $b$ is the duration correction factor; $n$ is the attenuation index; $T$ is the return period, years; $d$ is the duration, min.

The Chicago storms were designed as follows:

$$i(t) = \begin{cases} \dfrac{\frac{(1-n)(t_p - t)}{r} + b}{\left(\frac{t_p - t}{r}\right)^{n+1}} a(1 + c\lg T), & \text{if } t \le t_p \\[4mm] \dfrac{\frac{(1-n)(t - t_p)}{1-r} + b}{\left(\frac{t - t_p}{1-r}\right)^{n+1}} a(1 + c\lg T), & \text{if } t > t_p \end{cases} \tag{2}$$

where $i(t)$ is the average intensity at the $t$-th time-step, mm/min; $r$ is the time-to-peak coefficient, which is the ratio of the peak time ($t_p$, min) to the duration ($d$).

For this study, the values chosen are $T$ = 1, 2, 5; $r$ = 0.2, 0.4, or 0.6; $d$ = 120; $a$ = 16.715; $c$ = 1.1658; $b$ = 16.813; $n$ = 0.9302 [37]. Here are the reasons for each parameter setting: (1) $T$ = 1, 2, 5: Sponge cities typically design their LIDs to handle storms with short return periods (usually less than 5 years), as more intense storms exceeding this threshold can lead to overspending. (2) $r$ = 0.2, 0.4, 0.6: These values are based on local rainfall observations, where time-to-peak coefficients typically fall within the ranges of 0.1~0.2, 0.3~0.4, and 0.5~0.6. (3) $d$ = 120 min: This duration aligns with urban drainage system standards, which often focus on short-duration storms. Although a 180-min duration could be considered, we chose 120 min to emphasize infiltration characteristics. (4) $a$ = 16.715, $c$ = 1.1658, $b$ = 16.813, $n$ = 0.9302: These parameters are provided by the local weather bureau based on extensive, long-term rainfall observations.

Storm intensity is the primary factor influencing available water for infiltration, namely ponded surface water depth. Meanwhile, the temporal distribution of the Chicago storm is determined by Equation (2). Therefore, our analysis centers on the return period and time-to-peak coefficient.

*3.2. Storm Water Management Model*

SWMM stands out among urban stormwater models, rendering it the primary choice for this study. It offers an extensive suite of capabilities. These encompass infiltration simulation, surface runoff modeling, hydrological response assessment of LIDs, drainage network flow calculations, pollutant tracking, treatment evaluation, and overflow prediction [46]. SWMM is versatile, accommodating both single-event and long-term simulations, and excels in accurately modeling water dynamics within stormwater management [47]. Furthermore, its open-source nature allows for code redevelopment. Within SWMM, diverse methods are integrated to facilitate infiltration simulation, including the default Horton formula and the Green–Ampt method [48].

Table 1 provides an overview of the critical parameter values specifically adopted for porous pavements in SWMM. These values were derived from experiments and on-site observations and were provided by the Fengxi Management Committee [49]. These parameters are paramount in effectuating precise modeling and simulation of the infiltration and runoff processes within the catchment [50].

**Table 1.** Main parameter values of porous pavement in storm water management model.

| Layer | Parameter | Value | Layer | Parameter | Value |
|---|---|---|---|---|---|
| **Surface** | Berm height (mm) | 100 | **Storage** | Thickness (mm) | 500 |
| | Vegetation volume fraction | 0 | | Void ratio (voids/solids) | 0.75 |
| | Surface roughness | 0.01 | | Seepage rate (mm/h) | 1000 |
| | Surface slope (percent) | 0.5 | | Clogging factor | 0 |
| **Pavement** | Thickness (mm) | 60 | **Drain** | Flow coefficient | 0.5 |
| | Void ratio (voids/solids) | 0.15 | | Flow exponent | 0.5 |
| | Impervious surface fraction | 0 | | Offset (mm) | 0 |
| | Permeability (mm/h) | 1000 | | Open level (mm) | 6 |
| | Clogging factor | 0 | | Closed level (mm) | 0 |
| | Regeneration interval (days) | 0 | | Control curve | 0 |
| | Regeneration fraction | 0 | | | |

In pursuit of the most theoretically accurate outcomes, the dynamic wave model was deliberately chosen from among the routing models in SWMM. This model achieves heightened precision by solving the one-dimensional Saint Venant equations and adeptly replicates backwater flow effects by incorporating pipe storage, water return, import and export losses, and due consideration of countercurrent and pressure flow [51].

In each subcatchment, we derived most parameters through measurements or estimations based on underlying surface data and field investigations. These parameters encompass subcatchment area, imperviousness, slope, roughness, and facility sizes. Calibration of other SWMM parameters followed two criteria: first, minimizing errors in simulated outflow time series using Nash–Sutcliffe efficiency (NSE) [52], and second, minimizing errors in simulated peak flow rate using relative error. The parameters subjected to calibration [53] included subcatchment width, infiltration parameters, depression storage, and the percentage of runoff routed from impervious to pervious areas. It is important to note that the values of these parameters were constrained within limits recommended in the SWMM manual [39] and corroborated by relevant literature.

### 3.3. Horton Infiltration Method

#### 3.3.1. Governing Equations

The Horton formula has held a pivotal position within SWMM since its first release. Its classical form utilizes an exponential equation to calculate the reduction in infiltration capacity over time during rainfall events [54]:

$$f_p = f_\infty + (f_0 - f_\infty)e^{-k_d t} \tag{3}$$

where $t$ is the elapsed time (from the storm onset), h; $f_p$ is the infiltration capacity into the soil, mm/h; $f_\infty$ is the minimum (or equilibrium) value of $f_p$ at infinite time, mm/min; $f_0$ is the maximum (or initial) value of $f_p$ at the start of the storm, mm/h; $k_d$ is the decay coefficient, a constant reflecting how fast the infiltration rate decreases over time, 1/h. Soil conditions primarily influence the values of these parameters. Consequently, the actual infiltration rate ($f$) is determined as the lesser value between the infiltration capacity and actual storm intensity:

$$f(t) = \min\big[f_p(t), i(t)\big] \tag{4}$$

SWMM uses the integrated form to determine the cumulative infiltration capacity:

$$F(t_p) = \int_0^{t_p} f_p \mathrm{d}t = f_\infty t_p + \frac{(f_0 - f_\infty)}{k_d}(1 - e^{-k_d t_p}) \tag{5}$$

The actual cumulative infiltration ($F$) is calculated as follows:

$$F = f_\infty t_p + \frac{(f_0 - f_\infty)}{k_d}(1 - e^{-k_d t_p}) \tag{6}$$

Estimating the values of $f_0$, $f_\infty$, and $k_d$ for each subcatchment requires considering the physical properties of the soil and fitting the equation to multiple field or laboratory datasets from different sites. The value of $f_0$ is influenced by soil type, initial soil moisture content, and vegetation conditions, while $f_\infty$, the most sensitive parameter in the Horton method, corresponds to saturated hydraulic conductivity. The $k_d$ value depends on the soil's initial moisture content. Additionally, the recovery rate is not considered here due to the use of design storms with a duration of 120 min for the SWMM simulation.

### 3.3.2. Computational Scheme in Storm Water Management Model

The SWMM engine employs a computational scheme to calculate infiltration for the Horton method, as depicted in Figure 3 of Parnas et al. (2021) [55]. The process for determining the infiltration rate ($f$) in a subcatchment during a time step ($\Delta t$) under a storm is outlined as follows:

(1) Input the necessary variables, including rainfall rate ($i(t)$), ponded surface water depth ($d$), equivalent time ($t_p$) on the Horton curve, and constants $f_0$, $f_\infty$, and $k_d$.
(2) Calculate the available storm rate ($i_a$).
(3) If $i_a$ equals 0, update the current time ($t_p$) on the infiltration curve and set $f$ to 0. Otherwise, compute the cumulative infiltration volume using Equations (5) and (6) at times $t_p$ and $t_p + \Delta t$.
(4) Calculate the average infiltration rate over the time step.
(5) Update $t_p$ and update $f$ using Equation (4).

Subsequently, the following steps are performed for the catchment infiltration calculation for each time step within the SWMM engine:

(1) Determine if the area is pervious. If it is, apply the Horton formula to calculate the infiltration rate and volume for the time step. If it is not pervious, set the infiltration rate and volume to 0.
(2) Check for the presence of LIDs. If one exists, use the Green–Ampt model (allowing the consideration of surface ponding) to calculate the infiltration rate and volume for each LID facility. The infiltration volume of the subcatchment is obtained by summing the infiltration volumes of the pervious area and each LID facility.
(3) Compute the infiltration volume for the entire study area by summing up the infiltration volumes of each subcatchment.
(4) Determine the infiltration rate for the study area by dividing the infiltration volume of the study area by the area and time steps.

### 3.4. Field Investigation

The performance of the Horton model exhibits site-dependent behavior, closely linked to the soil textures prevalent in the monitoring sites. Soil infiltration monitoring was conducted at three distinct sites within WR8 in 2017, utilizing a portable double-ring infiltrometer to generate site infiltration curves [50]. The selected sites for monitoring included a lawn near Qinhuang Avenue, a wooded area near Xingxian Road, and a barren area near Tongxin Road, each representing distinct soil textures. Among the five infiltration models available in SWMM, the Horton model demonstrated superior fitting performance, as evidenced by its favorable performance across various evaluation metrics. This outcome underscores the suitability of the Horton model for characterizing infiltration dynamics at the WR8 site.

## 4. Results and Discussion

### 4.1. Area Changes after Adding Porous Pavements

Table 2 provides an overview of the areas allocated for schemes without porous pavements (no-PP scheme) and schemes with porous pavements (PP scheme). Our observations revealed that for each subcatchment, an increase of $n$ percent in the PP area resulted in proportional decreases of $np_1$ percent, $np_2$ percent, and $np_3$ percent in the impervious area without depression storage, impervious area with depression storage, and pervious area, respectively. $p_1$, $p_2$, and $p_3$ denote the percentages of the three underlying surfaces in the no-PP scheme.

**Table 2.** Areas ($m^2$) of subareas in subcatchments for no porous pavements (no-PP) and porous pavements (PP) schemes.

| Subcatchment | Area | No-PP Scheme | | | PP Scheme | | | |
|---|---|---|---|---|---|---|---|---|
| | | IA-NO [1] | IA | Pervious Area | IA-NO | IA | Pervious Area | Porous Pavements (% [2]) |
| s1 | 162,329 | 30,843 | 92,528 | 38,958 | 26,117 | 78,352 | 32,990 | 24,870 (15.3%) |
| s2 | 40,384 | 7673 | 23,019 | 9692 | 6373 | 19,120 | 8050 | 6841 (16.9%) |
| s3 | 21,504 | 4086 | 12,257 | 5161 | 3384 | 10,153 | 4275 | 3692 (17.2%) |
| s4 | 27,903 | 5302 | 15,905 | 6696 | 4861 | 14,583 | 6140 | 2319 (8.3%) |
| s5 | 14,799 | 2812 | 8435 | 3552 | 2233 | 6700 | 2821 | 3045 (20.6%) |
| s6 | 118,273 | 22,472 | 67,416 | 28,385 | 17,157 | 51,472 | 21,672 | 27,972 (23.7%) |
| s7 | 202,587 | 38,492 | 115,475 | 48,620 | 32,579 | 97,738 | 41,153 | 31,117 (15.4%) |
| s8 | 153,206 | 29,109 | 87,327 | 36,770 | 25,750 | 77,250 | 32,526 | 17,680 (11.5%) |
| s9 | 104,350 | 19,827 | 59,480 | 25,043 | 16,599 | 49,796 | 20,967 | 16,988 (16.3%) |

Notes: [1] IA-NO represents the impervious area with no depression storage; IA represents the impervious area with depression storage. [2] Percentage share, namely, the area ratio of porous pavements to the subcatchment.

### 4.2. Calibration Results of Storm Water Management

The SWMM model underwent calibration using data from three recorded storm events [38]. Table 3 lists the values of critical parameters for different subcatchments in SWMM. For the outflow series, the NSE values were 0.63, 0.84, and 0.76, while the relative errors for peak flow rates were 0.0038, 0.1552, and 0.0153 $m^3$/s, respectively. These results affirm the effectiveness of SWMM in accurately representing the hydrological processes within the study area. For additional information concerning the calibration and validation of SWMM, please refer to Section 3.3 of the study [38].

**Table 3.** Key parameters for different subcatchments in storm water management model.

| Parameter | Value |
|---|---|
| Width (m) | 121.7~450.1 |
| Slope (%) | 0.5 |
| Imperviousness (%) | 0.76 |
| Manning's $n$ for overland flow in impervious area | 0.013 |
| Manning's $n$ for overland flow in pervious area | 0.15 |
| Depression storage in impervious areas (mm) | 1 |
| Depression storage in pervious areas (mm) | 3.2 |
| Conduit roughness | 0.013 |
| Conduit diameter (m) | 0.6~2.2 |
| Conduit length (m) | 105.0~641.0 |
| Junction elevation (m) | 380.6~384.3 |
| Outfall elevation (m) | 380.5 |

The measured minimum infiltration capabilities at the three monitoring sites were 38.5, 94.4, and 118.6 mm/h at 10 degrees Celsius and 45.1, 110.4, and 138.8 mm/h at 20 degrees Celsius, respectively. These site-specific data were utilized for calibrating the SWMM model in conjunction with other observed storm-related data, such as storm and outflow time

series. Consequently, the catchment's initial infiltration capacity ($f_0$), minimum infiltration capacity ($f_\infty$), and decay constant ($k_d$) were estimated at 24.5 mm/h, 3.1 mm/h, and 5 h$^{-1}$, respectively.

The soils in WR8 predominantly consist of loamy clays, as mentioned in Section 2. Following the SWMM manual [40], clay and loam soil exhibit initial capacities of 25.4 and 76.2 mm/h (1 in/h and 3 in/h), respectively. The derived values for the Horton model align reasonably with those specified in the SWMM manual, thus providing further validation.

While this approach provides valuable information on local infiltration characteristics, it may capture a fraction of the spatial heterogeneity within the catchment under actual conditions. Therefore, future research should involve extensive investigations and monitoring at various locations, considering the diverse soil textures. Moreover, for improved calibration of the Horton model in SWMM, minimizing the bias in simulating runoff responses at point, subcatchment, and catchment scales using measured storm–runoff data at multiple sites requires further research to reduce uncertainty [36]. By adopting such a comprehensive approach, we can better elucidate the intricate dynamics of infiltration and enhance stormwater management strategies.

### 4.3. Uniform Storm Parameters Impact on Infiltration
#### 4.3.1. Catchment Scale

Figure 4 presents the catchment's infiltration capacities and intensities under uniform storms with 120 min and 1-, 2-, or 5-year return periods for porous pavements scheme. The three uniform storms (yellow bars) featured total depths of 20.7, 27.9, and 37.5 mm, respectively, accompanied by corresponding intensities of 0.1722, 0.2327, and 0.3126 mm/min. Notably, the depth and intensity of the 2-year (or 5-year) uniform storm were approximately 1.35 times (or 1.81 times) those of the 1-year uniform storm. Importantly, all three storm intensities remained below the maximum infiltration capacity of 0.4083 mm/min (equivalent to 24.5 mm/h).

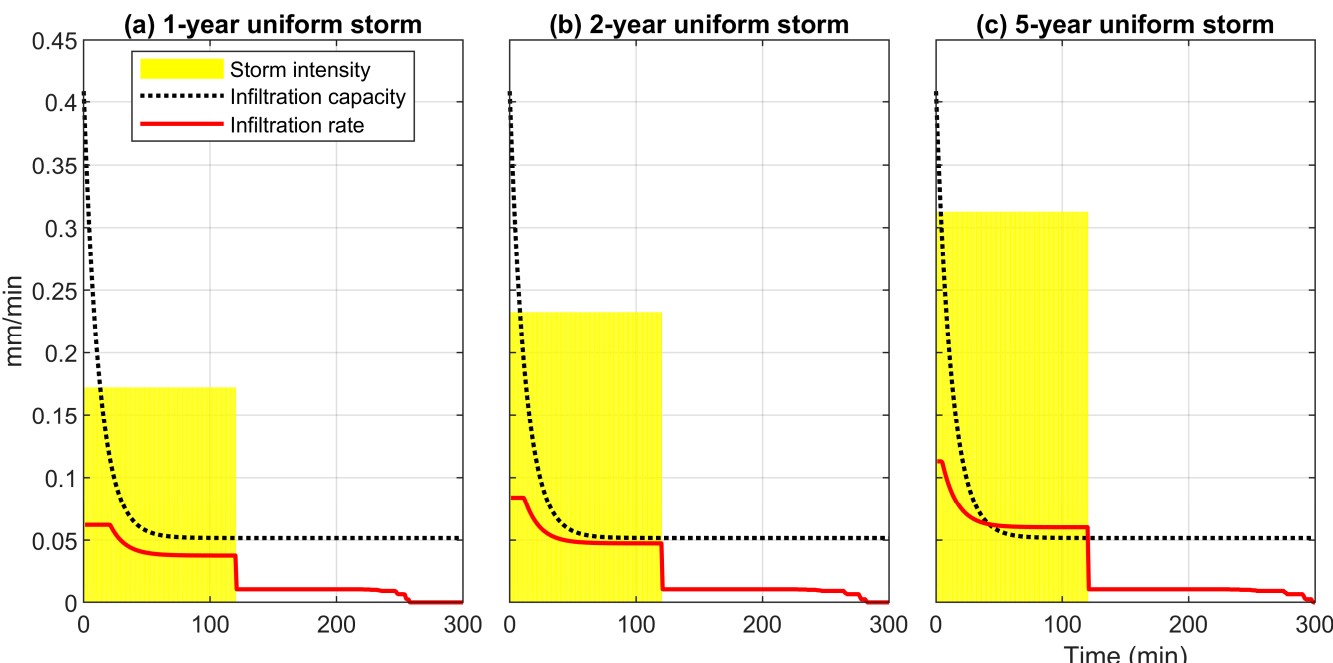

**Figure 4.** Catchment infiltration capacity and rate under uniform storms with 120 min duration and 1-, 2-, or 5-year return period for porous pavements scheme.

The catchment's infiltration capacities during the three uniform storms (black dotted lines in Figure 4) are theoretical values calculated using the Horton method, assuming sufficient water for infiltration. According to the Horton infiltration theory, when water

availability is limited, the actual infiltrability may be less than the infiltration capacity at a given time and for a specific soil. In other words, the infiltration process is either supply-controlled or profile-controlled. Remarkably, under each uniform storm, the process initially follows a supply-controlled pattern, transitioning to a profile-controlled state before returning to a supply-controlled mode.

The catchment's infiltration rates (red solid lines in Figure 4) offer the following insights: At the onset of each storm event, the infiltration rates remained constant (0.0622, 0.0840, and 0.1128 mm/min). However, the duration of this steady state was shorter under more intense storms; knee points were observed on the infiltration rate curves at 21, 11, and 4 min for the respective storms, indicating that the increased storm intensity led to a faster filling of soil pores during the initial stages of the infiltration process.

The catchment infiltration volume, determined by applying the definite integral method to the infiltration rate time series, is illustrated in Figure 5. A linear correlation emerged between the infiltration volume and the uniform storm intensity. Furthermore, a precise linear equation was derived to represent this relationship accurately. The observed pattern can be attributed to higher storm intensities resulting in larger infiltration rate time series, leading to greater infiltration volumes, as represented by the enclosed area under the infiltration rate curve. This finding aligns with the observation that cumulative infiltration exhibited significant variations [3].

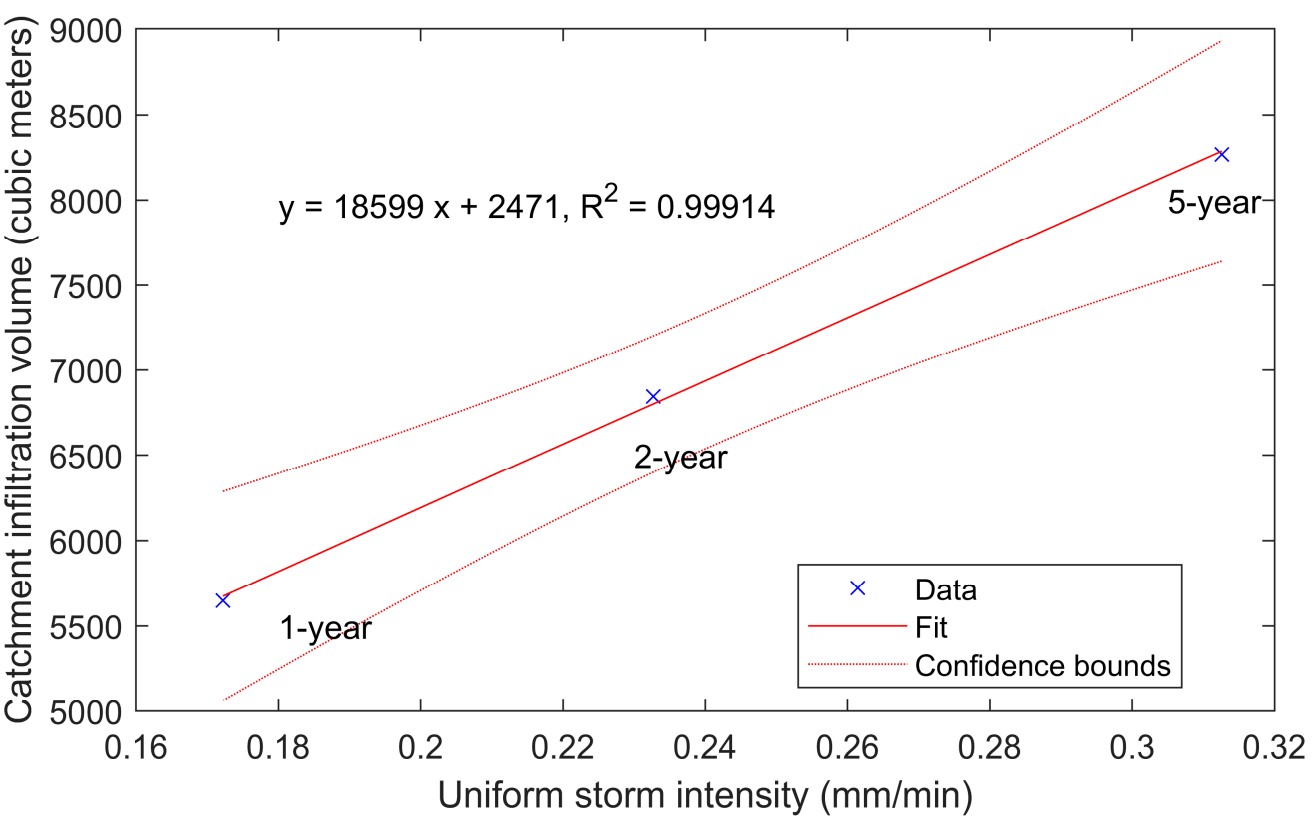

**Figure 5.** Catchment infiltration volumes under uniform storms with 120 min duration and 1-, 2-, or 5-year return period for porous pavements scheme.

### 4.3.2. Subcatchment Scale

The infiltration rates within subcatchments, including pervious areas and porous pavements, were analyzed under uniform storm conditions. Interestingly, the infiltration rates of pervious areas remained consistent within each subcatchment, regardless of whether porous pavements were present. Additionally, similar patterns in infiltration rates were observed across all subcatchments. To illustrate this, we present an example using

subcatchment s7, depicting the infiltration rates of its subareas under uniform storms with return periods of 1, 2, or 5 years for the porous pavements scheme, as shown in Figure 6.

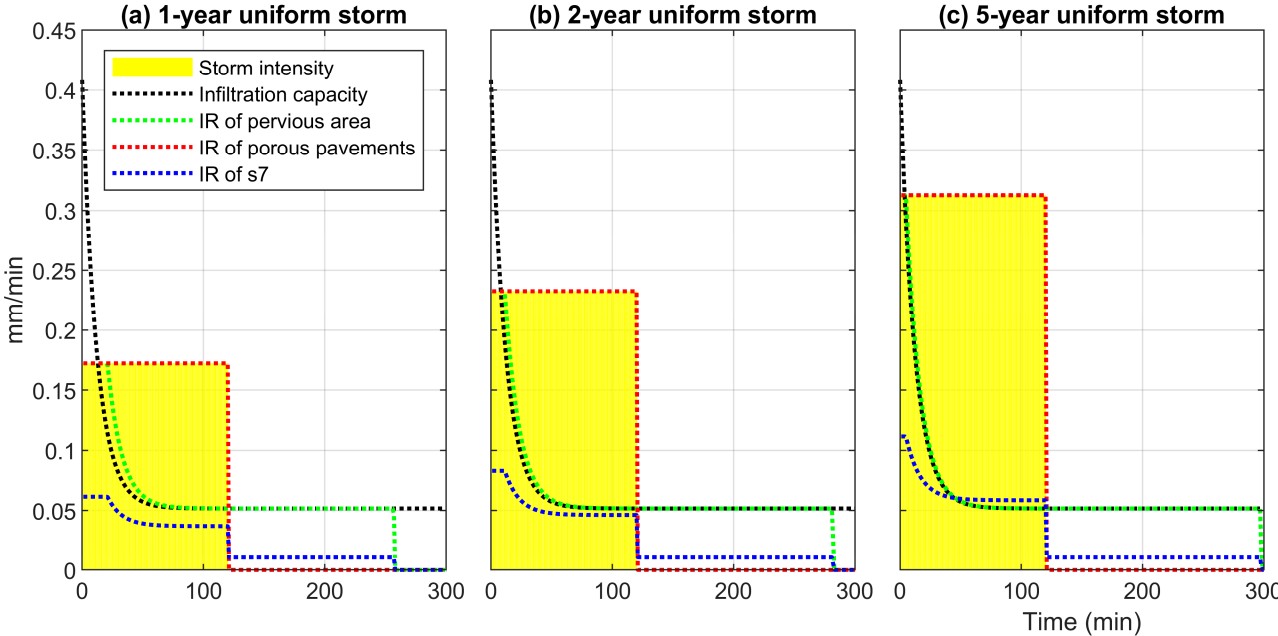

**Figure 6.** Infiltration rates (IRs) of pervious area, porous pavements, and subcatchment s7 under uniform storms with 120 min duration and 1-, 2-, or 5-year return period for porous pavements scheme.

The analysis of infiltration rates of pervious areas in subcatchment s7 under uniform storms (green dotted lines in Figure 6) revealed the following: (1) Initially, the infiltration rate remained constant, then gradually decreased, reached a state of equilibrium (equivalent to the minimum infiltration capacity), and eventually diminished to zero due to the absence of available rainwater for infiltration. (2) As the return period increased, the initial values of the infiltration rate rose and aligned with the corresponding storm intensities (0.1722, 0.2327, and 0.3126 mm/min). This behavior can be attributed to the Horton method, where the storm intensities were less than the maximum infiltration capacity (0.4083 mm/min, equivalent to 24.5 mm/h), resulting in infiltration rates equal to the storm intensities. (3) Higher storm intensities led to faster filling of soil pores, resulting in shorter durations (20, 11, or 4 min) of constant infiltration rates. (4) The actual infiltration rates may exceed the infiltration capacities on the Horton curve, as indicated by the green line surpassing the black line, as seen in Figure 6a, due to the initially inadequate amount of water available for infiltration.

Turning to the infiltration rates of porous pavements in subcatchment s7 under uniform storms (red dotted lines in Figure 6), it was evident that these rates remained constant throughout the storm and were equal to the storm intensities (0.1722, 0.2327, or 0.3126 mm/min). Subsequently, infiltration rates promptly dropped to zero upon the storm's cessation. This outcome can be attributed to porous pavements controlling the stormwater that falls on their surfaces and having sufficient infiltrability to filtrate the rainfall fully, thus aligning the infiltration rates with the storm intensities.

Moreover, the infiltration rates of subcatchment s7 (blue dotted lines in Figure 6) were examined, demonstrating similar patterns across different return periods, with higher return periods resulting in increased infiltration rates. Notably, the infiltration rates of s7 were significantly influenced by the infiltration rates of porous pavements compared with those of the pervious area. This finding underscores the impact of porous pavements on overall infiltration dynamics within the subcatchment.

Focusing on the infiltration rates of the catchment (red solid lines in Figure 4), pervious area (green dotted lines in Figure 6), and s7 (blue dotted lines in Figure 6), we observed

that the rate of descent in the infiltration rate curve increased with the higher storm return period. This pattern is consistent with the findings of Mu et al. [27], who reported that the infiltration rate curve became steeper with increasing rainfall intensity.

*4.4. Chicago Storm Parameter Impact on Infiltration*

4.4.1. Catchment Scale

　　Nine Chicago storms, each lasting 120 min and with return periods of 1, 2, or 5 years, and time-to-peak coefficients of 0.2, 0.4, or 0.6, were utilized to calculate the infiltration rates in SWMM. The catchment infiltrations are presented in Figure 7, revealing the following observations:

(1)　The infiltration rates (solid lines) peaked simultaneously with the Chicago storms. When the storm intensities exceeded the soil infiltrabilities, the infiltration rates equaled the infiltrabilities. However, at the onset of the storms, the soil infiltrability was not fully satisfied with low storm intensities, leading to gradual increases in the infiltration rates until they reached their maximum values during the storm peak.

(2)　The peak infiltration rate exhibits a weak positive correlation with the return period and a weak negative correlation with the time-to-peak coefficient. Specifically, under the Chicago storm with a time-to-peak coefficient of 0.2, 0.4, or 0.6, the correlation coefficients and $p$-values of the peak infiltration rate concerning the return period are 0.9814, 0.9816, or 0.9810 and 0.1230, 0.1222, or 0.1224, respectively. Conversely, under the Chicago storm with a return period of 1 year, 2 years, or 5 years, the correlation coefficients and $p$-values of the peak infiltration rate regarding the time-to-peak coefficient are −0.9550, −0.9384, or −0.9212 and 0.1918, 0.2247, or 0.2544, respectively. This can be attributed to storms peaking later, resulting in higher soil moisture content at the storm's peak, leading to reduced infiltration rates at that specific moment. However, all $p$-values exceed 0.05 (i.e., confidence level of 95%), indicating that the observed correlations lack statistical significance. Notably, the peak infiltration rates exhibited only minor changes, consistent with the findings of Fu et al. (2023), who reported that the maximum infiltration rate remained largely consistent [3].

(3)　The infiltration volumes were calculated, revealing a weak positive correlation with the return period. Specifically, under the Chicago storm with a time-to-peak coefficient of 0.2, 0.4, or 0.6, the correlation coefficients and $p$-values of infiltration volume regarding the return period are 0.9753, 0.9751, or 0.9747, and 0.1418, 0.1423, or 0.1434, respectively. Under the Chicago storm with a return period of 1 year, 2 years, or 5 years, the correlation coefficients and $p$-values of infiltration volume concerning the time-to-peak coefficient are −0.9350, −0.4647, or 0.7040, and 0.2307, 0.6923, or 0.5027, respectively. Significantly, these $p$-values exceed 0.05, indicating a lack of statistical significance in the observed correlations.

4.4.2. Subcatchment Scale

　　The infiltration rates of the pervious area under Chicago storms for the no-PP and PP schemes were identical. Thus, the infiltration processes in subcatchment s7 under Chicago storms with different return periods and peak-to-time coefficients were examined as an illustrative example. Figure 8 presents the infiltration rates of the pervious area, porous pavements, and subcatchment s7 under Chicago storms with 1-, 2-, or 5-year return periods and a peak-to-time coefficient of 0.4 for the porous pavement scheme.

　　The infiltration rates of the pervious area in s7 (green dotted lines in Figure 8) demonstrated that: (1) Initially, the infiltration rate increased and then decreased. During the early stages of the storms, the soil infiltrabilities exceeded the storm intensities, resulting in the infiltration rates being equal to the storm intensities. As the storm intensities increased and the infiltrabilities decreased, the infiltration rates peaked when these two values became equal. Subsequently, as the storm intensities continued to rise and surpass the infiltrabilities, the infiltration rates became equal to the infiltrabilities. As the infiltrabilities decreased

further, the infiltration rates equaled the minimum infiltration capacity until the water-input rates reached zero, resulting in an infiltration rate of zero. (2) With increasing return periods, the peak infiltration rates varied (0.2077, 0.1966, 0.1974 mm/min), and the timing occurred earlier (38, 34, 30 min). These peak infiltration moments were earlier than the peak storm (48 min). A non-significant negative correlation was observed between peak infiltration rates and the return period (correlation coefficient = −0.6426, *p*-value = 0.5557). Similarly, a weak negative correlation was identified between peak infiltration moments and the return period (correlation coefficient = −0.9608, *p*-value = 0.1789).

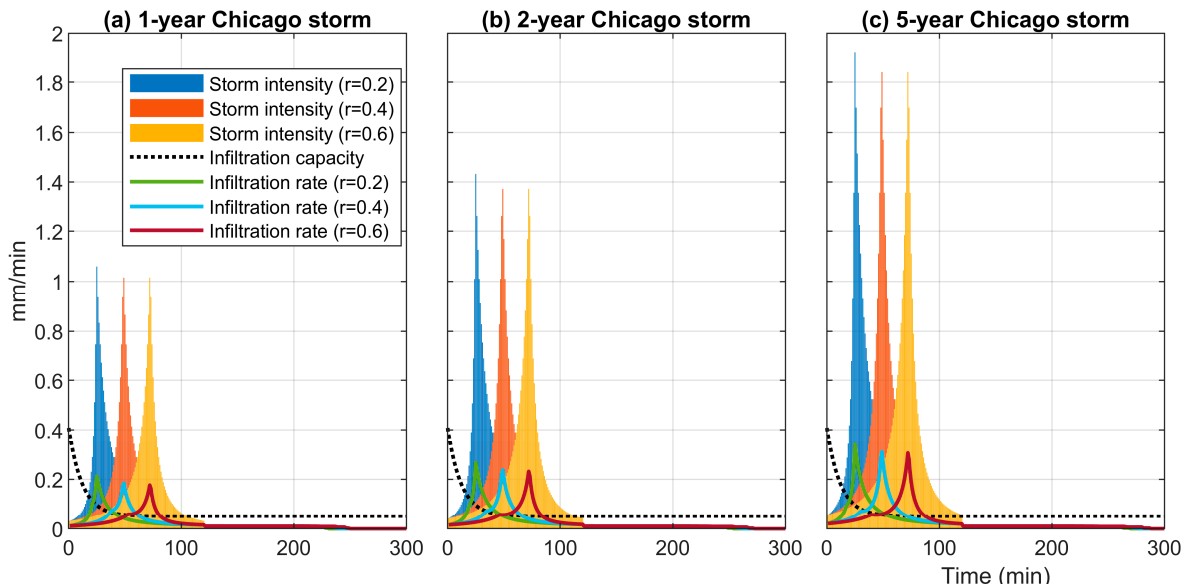

**Figure 7.** Catchment infiltration rates under Chicago storms with 120 min duration, 1-, 2-, or 5-year return period, and 0.2, 0.4, or 0.6 time-to-peak coefficient (denote as *r*) for porous pavement scheme.

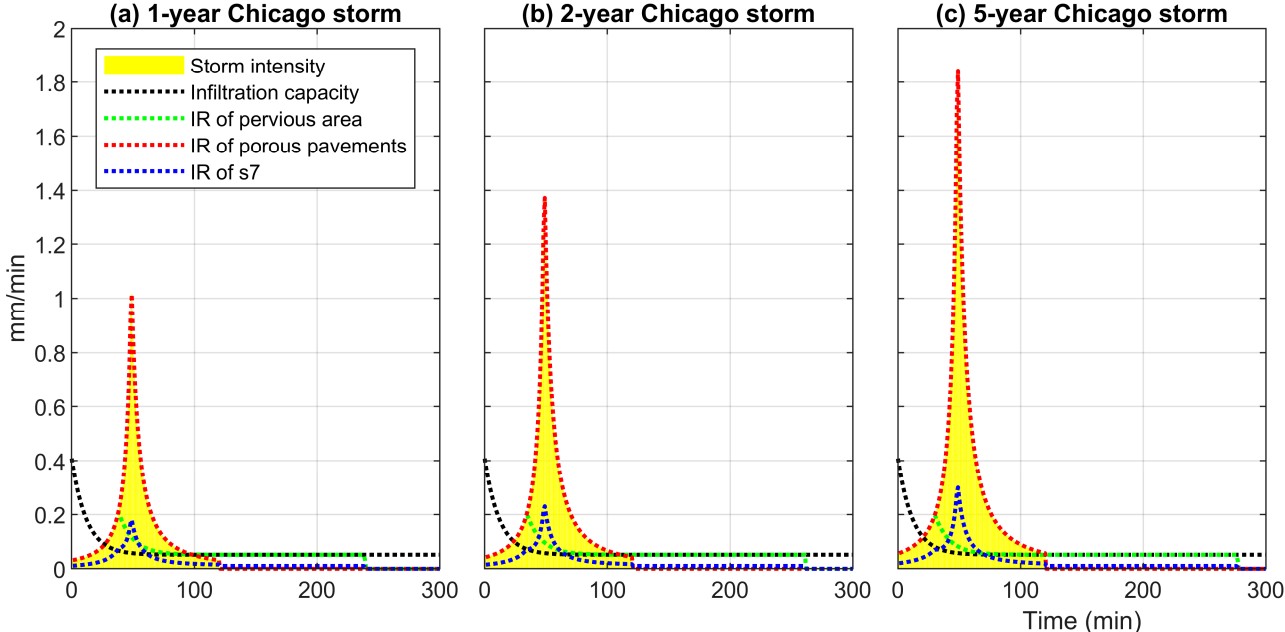

**Figure 8.** Infiltration rates (IR) of pervious area, porous pavements, and subcatchment s7 under Chicago storm with 120 min duration, 1-, 2-, or 5-year return period, and a time-to-peak coefficient of 0.4 for porous pavement scheme.

The infiltration rates of porous pavements in s7 (red dotted lines in Figure 8) provided the following insights: The infiltration rates equaled the storm intensities at any given time and immediately dropped to zero at the storm's end.

In addition, the infiltration rates of subcatchment s7 (blue dotted lines) generally followed the patterns of storm intensities, initially increasing and then decreasing. They may exceeded those of the pervious area when the infiltration rates of porous pavements were significant, resulting in larger catchment infiltration rates after area-weighted averaging. As the return period increased, the peak infiltration rates varied (0.1794, 0.2303, 0.2998 mm/min). A weak positive correlation was identified between the peak infiltration rates and the return period (correlation coefficient = 0.9817, *p*-value = 0.12).

Figure 9 illustrates the infiltration rates of the pervious area, porous pavements, and subcatchment s7 under Chicago storms, with a duration of 120 min, a 5-year return period, and time-to-peak coefficients of 0.2, 0.4, or 0.6 for the porous pavement scheme. The 5-year return period was chosen for analysis because it exhibited similar patterns to other return periods.

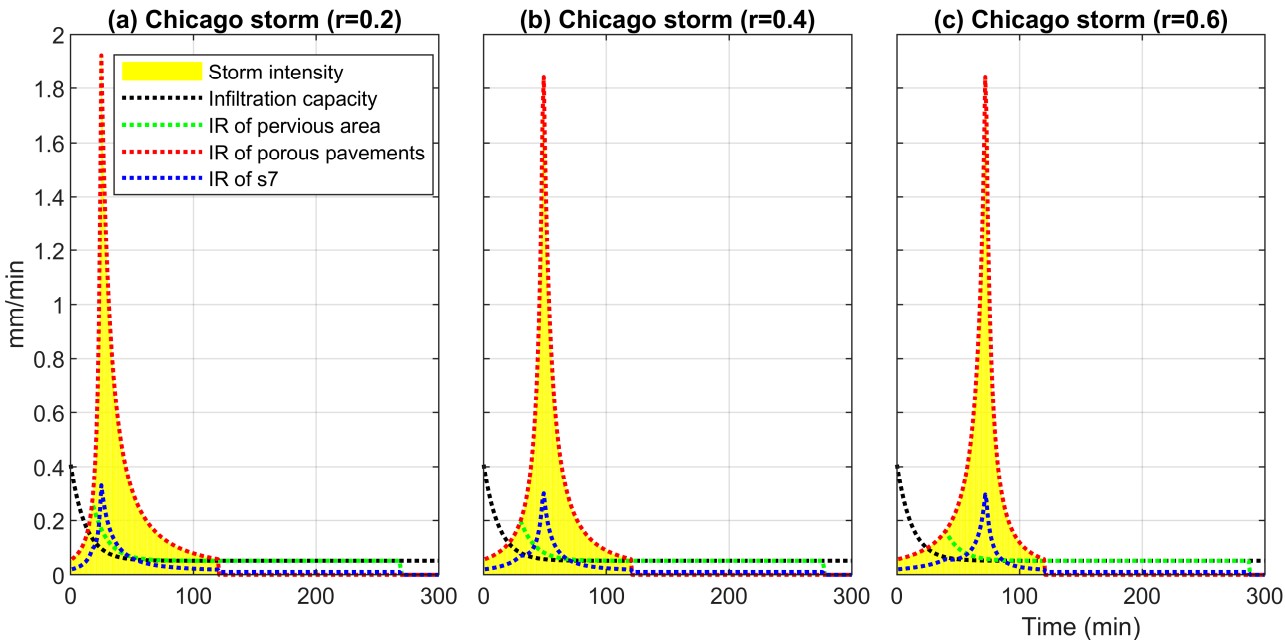

**Figure 9.** Infiltration rates (IR) of pervious area, porous pavements, and subcatchment s7 under Chicago storm with 120 min duration, 5-year return period, and 0.2, 0.4, or 0.6 time-to-peak coefficient (denote as r) for porous pavement scheme.

Our findings regarding the infiltrations of the pervious area (green dotted lines) reveal that a larger time-to-peak coefficient (0.2, 0.4, 0.6) resulted in a smaller peak infiltration rate (0.2593, 0.1974, 0.1585 mm/min) and a longer time to reach the peak infiltration rate (18, 30, 40 min). This observation can be attributed to the fact that with a larger time-to-peak coefficient, the cumulative precipitation at any given time before 72 min (time-to-peak coefficient of 0.6) was smaller, leading to smaller soil water content. A weak negative correlation was observed between the peak infiltration rates and the time-to-peak coefficient (correlation coefficient = −0.9914, *p*-value = 0.08). Conversely, there is a positive correlation between the peak infiltration moments and the time-to-peak coefficient (correlation coefficient = 0.9986, *p*-value = 0.0334).

On the other hand, the infiltration rates of subcatchment s7 (blue dotted lines) reached their peaks (0.3398, 0.3099, 0.3053 mm/min) at 25, 49, and 72 min, respectively. The peak moments of infiltrations aligned with or were close to those of the storms. There is a weak negative correlation between the peak infiltration rates and time-to-peak coefficient (correlation coefficient = −0.9220 and *p*-value = 0.25).

The Horton Model is sensitive to rainfall intensity throughout the simulation, and storm intensity and temporal distribution are crucial for accurate runoff prediction [36]. Our findings show that parameters like the Chicago storm's return periods and time-to-peak coefficient significantly impact infiltration simulation results in SWMM for sponge cities. These observations support previous research by [35].

## 5. Conclusions

This research examined storm parameters' impact on infiltration within a sponge city, particularly the return period and time-to-peak coefficient. Within the SWMM framework, the Horton and Green–Ampt infiltration models were employed for the pervious areas of subcatchments and porous pavements, respectively. We concluded that:

(1) Increasing the area of porous pavements results in proportional reductions in the impervious area without depression storage, the impervious area with depression storage, and the pervious area based on their initial area ratios. The infiltration rates of porous pavements under uniform and Chicago storms were supply-controlled.

(2) The infiltration rates of the pervious areas, subcatchments, and catchment under uniform storms exhibit a consistent initial stage, with the duration of this steady state becoming shorter as the return period increases. The catchment infiltration volumes demonstrate a linear growth trend with higher uniform storm intensities.

(3) The peak infiltration rate within pervious areas exhibits a non-significant negative correlation with the return period, while those within subcatchments and the overall catchment display non-significant positive correlations with the return period. The peak infiltration rate for pervious areas, subcatchments, and the catchment demonstrates non-significant negative correlations with the time-to-peak coefficient.

(4) The peak infiltration moments within pervious areas show non-significant negative correlations with the return period and non-significant positive correlations with the time-to-peak coefficient. Infiltration rates of porous pavements, subcatchments, and the overall catchment peak simultaneously to Chicago storms.

Our findings significantly advance the understanding and prediction of soil infiltration rates within sponge cities. Notably, our results underscore the critical importance of integrating considerations related to return periods and time-to-peak coefficients into infiltration analyses and the planning of infiltration-based facilities. We strongly recommend the implementation of porous pavements alongside impervious surfaces to facilitate the infiltration of runoff. It is imperative to recognize the diverse infiltration patterns that manifest under different storm scenarios, as they should inform the adaptive design, planning, and management of porous pavements. The effectiveness of these systems is substantially influenced by the characteristics of the rainfall events they encounter. Therefore, optimizing porous pavement locations and properties should be tailored to the local rainfall characteristics. Furthermore, it is worth noting that porous pavements exhibit enhanced performance when dealing with rainfall events characterized by larger time-to-peak coefficients. Consequently, retention-based solutions should be emphasized as an alternative strategy to mitigate the impacts of such rainfall events.

Nonetheless, it is crucial to recognize the limitations of our study. Specifically, despite its widespread application, the Horton model does not account for the effect of cumulative water layer depth on infiltration intensity, a consideration addressed by the Green–Ampt model. Additional advances are required for greater applicability, especially facilitating continuous simulations that include ponding and non-ponding conditions. Also, we advocate examining the dynamic connections between soil properties, storm events, runoff dynamics, and the effect of vegetation coverage in light of future research priorities. Furthermore, invaluable would be an investigation of the Hortonian overland flow mechanism and extensive field measurements to investigate the spatiotemporal heterogeneity of infiltration capacity and intensity across the catchment. These future attempts are anticipated to yield a thorough understanding of infiltration mechanisms, enabling the design of sustainable ur-

ban infrastructure that effectively manages stormwater, reduces flood risks, and encourages water conservation.

**Author Contributions:** Conceptualization, Y.Y.; methodology, Z.S. and Y.Y.; software, Z.S. and Y.Y.; validation, Y.Y. and X.X.; formal analysis, Z.S. and Y.Y.; investigation, Z.S. and Y.Y.; resources, Y.Y.; data curation, Z.S.; writing—original draft preparation, Y.Y. and Z.S.; writing—review and editing, Y.Y.; visualization, Z.S. and Y.Y.; supervision, Y.Y. and D.L.; project administration, Y.Y.; funding acquisition, Y.Y. All authors have read and agreed to the published version of the manuscript.

**Funding:** This research was funded by the National Natural Science Foundation of China, grant number 52009099 and 52279025. The APC was funded by the China Postdoctoral Science Foundation Funded Project (2019M653882XB) and the Joint Institute of the Internet of Water and Digital Water Governance (sklhse-2019-Iow06).

**Data Availability Statement:** Not applicable.

**Acknowledgments:** We thank the editors and reviewers for their helpful comments and suggestions.

**Conflicts of Interest:** The authors declare no conflict of interest.

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
