# Peer review of "Impact of Storm Characteristics on Infiltration Dynamics in Sponge Cities Using SWMM"

_water, doi:10.3390/w15193367_

Round 1

Reviewer 1 Report (New Reviewer)

Author Response

Reviewer 2 Report (New Reviewer)

The topic and objective of the study are relevant and important in current situations.  Overall, the manuscript is acceptable. Sufficient information about previous study findings is provided. The methods that are used to analyze the data are appropriate. However, I have the following suggestions:

·         Provide more context or rationale for the choice of specific storm parameters in the Chicago storm analysis. Explain why the chosen parameters are relevant and how they contribute to the study's objectives.

·       Consider expanding the conclusion by discussing potential applications of the study's findings. How can the insights gained from this research inform stormwater management strategies in real-world urban environments? Are there any specific recommendations or next steps based on the findings?

There are instances where the writing could be further improved for clarity. Some sentences are quite lengthy and complex. Consider breaking them down into smaller sentences to improve readability.

Author Response

Reviewer 3 Report (New Reviewer)

please see the attached file

The paper is well written

Author Response

Reviewer 4 Report (New Reviewer)

Review

Title: Impact of Storm Characteristics on Infiltration Dynamics in Sponge Cities using SWMM

No. of manuscript 2571936

The article concerns a very current issue, which is the management of rainwater and the introduction of LID forms to cities. The article requires very thorough additions and corrections.

1)      The most important consideration is the issue of calibration. It is vague and insufficiently explained to such an extent that one may have doubts as to whether it has been performed at all. The article talks about model calibration, but in the context of entering data into the model. And that's not what calibration is about. The results of the model should be compared with the standard - field research or the results of mathematical calculations. Here, the authors compare the field research with the ranges given in the SWMM Manual regarding the data INSERT to the model. How was the calibration done? A reference to another article on this issue is not sufficient.

2)      2) The question of infiltration. The reason for choosing infiltration method : "is simple to comprehend and implement" - is not the best way. First of all, the method should be well adjusted to the research field/research region. The introduction contains information about the Horton method, but there is no information about the Green-Ampta method. It is unclear why the whole article gives the impression that only Horton's method is modeled, while in 2 places suddenly information about using also Green Ampt's method appears. In addition, the article lacks a clear statement that the porous pavement is modeled as one of the forms of LID, and in turn indicates that the Green-Ampta method is used for LID. In fact, these porous pavements are the main reason for writing the article. It would also be worth adding why the methods: modified Horton and modified Green - Ampt were not used, because they are also implemented in SWMM...

3)       Input parameters to the SWMM model. The adoption of individual parameters for simulation is also not clear in every case. There is no information characterizing individual catchments, as well as the conduit - hence Figure 1 does not provide too much information. When creating rain data, no explanation as to why this particular input was taken? For instance 120 minutes? Reference to item [38] is insufficient. Not even 1 sentence of explanation is missing. In Chicago storm - the same - why r was this value adopted? Isn't a porous pavement classified as a pervious area? For example, for s1, if the PP area is 24871, is there a PParea within the pervious area? Especially since the PP area is modeled as a permeable pavement. This should be marked and demarcated (LID and pervious area). You should add information about possible overwriting of certain parameters by the SWMM program. Not every reader is an SWMM expert, and these are important aspects. Was LID introduced into existing catchments or were separate catchments created for this form of LID? Does this need elaboration? If they were created within the catchment, was the width of the catchment changed? Also, has initially saturated information been entered for the LID? What? Table 1 - on what basis were these values adopted? There is no information on this subject (and it is mentioned in the text that they are crucial).

4)      Some graphics are not fully legible and raise doubts. Figure 1: the caption is inadequate to the drawing - the description does not mention the diagram from SWMM, which is presented in the aerial pohtogrpah. Figure 2 - profile (b) contains completely unclear references, no explanation in the text, too large mental abbreviation, unacceptable in this context. Figure 3 does not contain a reference to literature, while there is a reference to literature in the text. And here again the lack of clarity...... Also, is it to be understood from this diagram that the calibration is only about evaluating whether reasonable results have been obtained? This is a bit too little for further such serious inference that is contained in the article. Figure 5 - it is advisable to provide information on the significance of the correlation result and the adopted confidence level.

5)      The article talks about positive and negative correlation. Whenever correlation results are reported, information on significance tests is required, and the value of the correlation coefficient should also be provided. Especially since the article previously even mentions R2. No consistency in the presentation of results.

6)      Table 2 should be constructed in such a way that it can be checked that the sum of individual shares gives a value equal to the total area. Requires reorganization. In addition, there are calculation errors - the sum does not always give the value as for the catchment area and these are not differences by one unit due to rounding. Example - row no. 1.

7)       In addition, it is required to provide a detailed characteristics of the catchment area with regard to the types of surfaces that occur. Here it is stated in an unclear and unclear way. There is no information on the percentage share of newly introduced forms of LID (i.e. porous pavement) in individual catchments - data from Table 2 need to be supplemented. There is no information about this here.

8)      Other remarks. An average annual precipitation – for what period of years is this annual average? Data on precipitation are laconic and even the period from which the annual data is averaged is not given. Line 14 – 15/ “Increasing the area of porous pavements leads to proportional reductions in areas of subareas within each subcatchment” - unclear, what do you mean by: areas of subareas within each subcatchment;

Ok.

Round 2

Reviewer 1 Report (New Reviewer)

Thank you for your cooperation. I accept replies and corrections throughout this work.

Crash on line 444: "0.2, 0.4 or 0.8" (- value 0.8?) and on lines 712+715 (- literature?).

This manuscript is a resubmission of an earlier submission. The following is a list of the peer review reports and author responses from that submission.

Round 1

Reviewer 1 Report

In this study, SWMM model was used to simulate the change of infiltration amount according to rainfall characteristics. 

The results of this study are not an exploration of the actual natural phenomenon, but an analysis of the characteristics of the model. 

The same process as this study should be carried out with observed rainfall and observed infiltration data. 

Reviewer 2 Report

The topic of the paper under the title “Impact of Storm Characteristics on Infiltration Dynamics in Sponge Cities using SWMM” is interesting and within the scope of the Water journal. However, some issues require clarification and improvement. Therefore, I recommend major revisions before the publication of the manuscript. Please find the details below.

1. The authors have indicated (lines 50-52) that most studies identify the Horton infiltration model as suitable for use. However, the use of this model is often due to the fact that it is automatically applied in SWMM, and additionally relatively easy to describe. This model does not take into account the influence of the height of the layer of water accumulated on the surface on the intensity of infiltration, while, for example, the Green-Ampt model considers this variable. Please better justify the choice of the infiltration model in the text, referring to publications that also use other models to map the rainwater infiltration process, both within the catchment area and stormwater infiltration facilities. Please also avoid using lumped references. Reference to six items of literature in one place means that the reader does not know exactly what information can be found in individual papers.

2. Please describe the study area in more detail in section 2. Please add information about climatic conditions and soil type in the study area. The infiltration intensity values indicated in the text indicate that these are rather soils with a low infiltration capacity, but this requires explanation. Please add website addresses to the list of References and cite them in the text using numbers in square brackets.

3. In lines 115-117 the authors indicate that they used “the kinematic wave model for conveyance calculation”. Please justify the choice of this method in the text. The dynamic wave model better reflects the functioning of rainwater management systems.

4. Figure 3 suggests that one of the stages of the analysis was the calibration and validation of the SWMM model. Please add more information on this. What rainfall data was used for this purpose? What indicators were used to evaluate the process?

5. My main concern is to assume a rainfall duration of 120 min. Why was this duration chosen? What about rainfalls of other lengths? As the duration of rainfall changes, so does the intensity. In the opinion of this reviewer, analyzing three return periods for rainfall of constant length is not enough to assess the infiltration process.

6. Another important problem is the lack of discussion. The authors cite 40 items in the Introduction. The remaining 9 are in sections 2 and 3. In the section entitled "Results and Discussion", the reviewer did not find any reference to the literature. Please discuss your findings in relation to other authors' findings. Please also describe the limitations of the study, as this is currently missing from the manuscript.

Please also consider the following issues:

- Figure 1 – Please explain what is in figures a-d?

- Figure 2 – Shouldn't the descriptions of figures a and b be interchanged?

- Figure 2 – Please change “Permeabilit” to “Permeability”

- Wouldn't it be better to write equations (5-7) as two equations?

- Figure 5 – Please check the caption. The phrase “for porous pavement” is duplicated

Best regards